# Non-Coding RNAs in Breast Cancer: Intracellular and Intercellular Communication

**DOI:** 10.3390/ncrna4040040

**Published:** 2018-12-12

**Authors:** Carolyn M. Klinge

**Affiliations:** Department of Biochemistry & Molecular Genetics, University of Louisville School of Medicine, Louisville, KY 40292, USA; carolyn.klinge@louisville.edu; Tel.: +502-852-3668

**Keywords:** ncRNA, miRNA, lncRNA, breast cancer, transcription, estrogen receptor, resistance, cancer stem cells

## Abstract

Non-coding RNAs (ncRNAs) are regulators of intracellular and intercellular signaling in breast cancer. ncRNAs modulate intracellular signaling to control diverse cellular processes, including levels and activity of estrogen receptor α (ERα), proliferation, invasion, migration, apoptosis, and stemness. In addition, ncRNAs can be packaged into exosomes to provide intercellular communication by the transmission of microRNAs (miRNAs) and long non-coding RNAs (lncRNAs) to cells locally or systemically. This review provides an overview of the biogenesis and roles of ncRNAs: small nucleolar RNA (snRNA), circular RNAs (circRNAs), PIWI-interacting RNAs (piRNAs), miRNAs, and lncRNAs in breast cancer. Since more is known about the miRNAs and lncRNAs that are expressed in breast tumors, their established targets as oncogenic drivers and tumor suppressors will be reviewed. The focus is on miRNAs and lncRNAs identified in breast tumors, since a number of ncRNAs identified in breast cancer cells are not dysregulated in breast tumors. The identity and putative function of selected lncRNAs increased: nuclear paraspeckle assembly transcript 1 (*NEAT1*), metastasis-associated lung adenocarcinoma transcript 1 (*MALAT1*), steroid receptor RNA activator 1 (*SRA1*), colon cancer associated transcript 2 (*CCAT2*), colorectal neoplasia differentially expressed (*CRNDE*), myocardial infarction associated transcript (*MIAT*), and long intergenic non-protein coding RNA, Regulator of Reprogramming (*LINC-ROR*); and decreased levels of maternally-expressed 3 (*MEG3*) in breast tumors have been observed as well. miRNAs and lncRNAs are considered targets of therapeutic intervention in breast cancer, but further work is needed to bring the promise of regulating their activities to clinical use.

## 1. Introduction

Breast cancer is the most commonly diagnosed cancer and second leading cause of cancer death among women in the United States (U.S.). Breast tumors are heterogeneous, and are pathologically classified according to their expression of key proteins by immunohistochemical (IHC) staining (at a minimum 1% [1]): estrogen receptor α (ERα), gene *ESR1*, tumors are termed ER positive (ER+), progesterone receptor (PR, gene *PGR*), and human epidermal growth factor receptor 2 (HER2, gene *ERBB2*). Tumors that lack these three protein markers are “basal-like” and referred to as triple negative breast cancer (TNBC). Most primary breast tumors are ER+/PR+/HER2-, and patients are treated with surgery, radiation, and endocrine therapies (also referred to as antiestrogen therapies) Endocrine therapies employ aromatase inhibitors (AI), e.g., letrozole, to block the conversion of androgens to estrogens, or tamoxifen (TAM), which is a selective ER modulator (SERM) that competes with estrogens, including estradiol (E_2_) for binding ER. Most postmenopausal women with ER+ breast tumors receive AI therapy, while American Society of Clinical Oncology (ASCO) guidelines recommend 10 years of TAM for premenopausal women [2]. Unfortunately, 30–40% of patients develop resistance to endocrine therapies and develop metastatic disease [3,4]. Multiple mechanisms are involved in acquired endocrine resistant breast cancer [5]. Approximately 25–40% of metastatic tumors in breast cancer patients treated with AIs show *ESR1* mutations within the ligand binding domain (LBD) [6]. These mutations result in the ligand-independent transcriptional activity of the mutant ERα and reduce the efficacy of ER antagonists, including the selective ER downregulators (SERDs) fulvestrant, AZD9496, RU-58688, and GDC-0810 [7]. Both *ESR1* Y537S and D538G mutations are associated with more aggressive disease biology and shorter survival [8]. Recent clinical data suggest that the addition of the CDK4/6 inhibitor palbociclib in combination with letrozole provides a benefit in advanced disease [9,10,11,12]. Gene expression analysis in primary breast tumors for 50 genes in the PAM50 test has allowed the further dissection of molecular phenotypes that have clinical implications for individualizing patient treatment [13]. As of yet, no non-coding RNAs (ncRNAs) are used in commercial diagnostic tests; however, there is great interest in identifying circulating microRNAs (miRNAs) [14,15,16] and long non-coding RNAs (lncRNAs) [17,18,19] in breast cancer diagnosis and for monitoring therapeutic response.

Over the past two decades, we have learned that 99% of the total cellular RNA content of human cells consists of ncRNAs that are classified by size and function [20]. Transfer RNA (tRNA) (89%) and ribosomal RNA (rRNA) (8.9%) constitute the majority of ncRNAs, followed in abundance by messenger RNAs (mRNAs) (0.9%). Thus, the remaining ncRNAs, including circular RNA (circRNA), small nuclear RNA (snRNA), small nucleolar RNA (snoRNA), miRNA, and lncRNA together account for ~1% of total ncRNA. Despite their low abundance, these ncRNAs play critical roles in transcription, post-transcriptional processing, and translation [21]. In addition, because ncRNA can be packaged into extracellular vesicles (EV), including exosomes [22], they provide a mechanism for intercellular communication by the transfer of miRNA and lncRNA to recipient cells both locally and systemically [23]. It is important to note that the levels of ncRNA expression, their post-transcriptional modification (particularly lncRNAs), and their subcellular distribution are important to consider in assigning their potential function [24]. As concluded by Palazzo and Lee, it is ultimately critical to examine the biological function of each identified ncRNA on a case-by-case basis [24]. Table 1 lists the best-characterized regulatory ncRNAs with roles in breast cancer, as well as their sizes and function.

LncRNAs and miRNAs are considered to be epigenetic regulators of human cancers [25]. Both pri-miRNAs and lncRNAs are post-transcriptionally modified, e.g., by the methylation on N6 of adenosine (m^6^A), which alters their activities, providing an additional layer of regulation termed “epitranscriptomics” [26]. In addition, there is one report that a pool of inactive miR-34 lacking the 5′-phosphate in cancer cells, including MCF-7 cells, is activated by 5′-end phosphorylation by ATM serine/threonine kinase (ATM) and cleavage and polyadenylation factor I subunit 1 (CLP1) in response to DNA damage, in turn allowing AGO2 loading [27]. At least three times more reports are listed in PubMed on miRNAs than lncRNAs in endocrine cancers [28].

## 2. Small Nuclear RNAs and Small Nucleolar RNAs in Breast Cancer

snRNAs and snoRNAs play roles in mRNA splicing and rRNA maturation, respectively (reviewed in [67]). Elevated snoRNA biogenesis is required for the high rRNA expression that is needed for ribosomal biogenesis and enhanced protein synthesis in breast cancer (reviewed in [68]). Some snoRNAs are expressed in a cell-specific manner, including SNORD46 and SNORD42A in mammary glands and lymphoblastoid cells in the ENCODE (Encyclopedia of DNA elements) dataset (reviewed in [69]). Thirteen snoRNAs were identified as having prognostic relevance in breast cancer, including the downregulation of SNORD46 and SNORD89 [70]. The clonal evolution of successful cancer cells is guided by intracellular and intercellular mutations that provide advantages or disadvantages to cell lineages. This evolution is mediated in part by ribosomal alterations that are driven by mutations in uL18/RPL5 ribosomal protein genes in ~34% of breast tumors [71]. Many snoRNAs are processed into stable miRNA-like oligomers that are called ‘small nucleolar RNA-derived RNAs’ (sdRNAs) [72]. sdRNA-93 was reported to play a role in MDA-MB-231 cell invasiveness, and is overexpressed in Luminal B/HER2+ breast tumors relative to other types of breast tumors or normal breast tissue [73]. Other studies show that alterations in snoRNAs may promote carcinogenesis and favor caner stem cell phenotypes [71].

## 3. Circular RNAs

During the splicing of exons and removal of introns from heterogeneous nuclear RNA (hnRNA) to generate mRNA, “backsplicing” can generate circular RNAs (circRNAs) that are usually exported to the cytoplasm. CircRNAs are considered stable, although they are expressed at <10% of their associated linear mRNA transcripts (reviewed in [74]). circRNAs are evolutionarily conserved, suggesting that they are critical for cellular activities. Several factors regulate circRNA levels, including exon-skipping events, complementary sequences of flanking introns, RNA-binding proteins, and the amount of more than 25 of the ~170 proteins that constitute the spliceosome [74]. circRNAs can act as competing endogenous RNA (ceRNA, also called ‘sponge’) for microRNAs, thus reducing miRNA repression of their target mRNAs [75]. circRNAs modulate the stability of mRNAs, interact with RNA binding proteins, and regulate gene transcription by interaction with RNA pol II [29]. In addition, recent studies show that some circRNAs are translated, including circ-SHPRH, which generates a 17 kDa SHPRH-146aa protein that acts as a novel tumor suppressor protein and a protective decoy for its full-length SHPRH protein in glioblastoma [76,77].

At least six circRNAs have been reported as having roles in breast cancer (Table 1). Examples of circRNAs that are increased in breast tumors and classified as having oncogenic activity in breast cancer cell lines by acting as ceRNA for miRNAs include: hsa_circ_ABCB10 (ceRNA for miR-1271 [34]); hsa_circ_0011946 (ceRNA for miR-26a/b and thus upregulating RFC3 in MCF-7 cells) [78], hsa_circ_0052112 (ceRNA for miR-125a-5p in MDA-MB-231 cells) [79], and hsa_circ_0008039 (ceRNA for miR-432-5p and thus increasing E2F3 expression in MCF-7 and BT-20 cells) [38]. Recently, hsa_circ_0007294 (circANKS1B) was shown to be upregulated in TNBC tumors compared with all other breast tumor types, and circANKS1B acts as a ceRNA for miR-148a-3p and miR-152-3p, resulting in the increased expression of USF1, which increases transforming growth factor β1 (TGF-β1) expression to stimulate epithelial-to-mesenchymal transformation (EMT) [80]. It is likely that the further examination of RNA profiling data will identify more circRNAs in breast tumors, which will necessitate further molecular and functional studies about their roles in tumor formation, progression, and metastasis.

## 4. PIWI-Interacting RNAs

piRNAs work in PIWI–piRNA complexes to regulate gene expression at the epigenetic and post-transcriptional levels. piRNAs are the guardians of genome integrity by epigenetically silencing transposable elements by DNA methylation [81]. PIWI proteins are aberrantly expressed in breast cancer, including a high expression of *PIWIL2*, which is a member of the PIWI/AGO gene family [82], and *PIWIL4* is highly expressed in TNBC tumors and cell lines, and correlates with distant metastasis fatality [83]. Experiments in MDA-MB-231 TNBC cells demonstrated that *PIWIL4* activates TGF-β, MAPK/ERK, and fibroblast growth factor (FGF) signaling, and suppresses the expression of MHC class II genes [83]. The analysis of existing small RNAs in–RNA seq data identified >100 piRNAs in breast cancer cells, some of which were differentially expressed in MCF-7, ZR-75.1, and SKBR3 breast cancer cells relative to MCF-10A immortalized ‘normal’ breast epithelial cells [41]. The role of piRNAs in breast tumors remains to be thoroughly examined.

## 5. MicroRNAs in Breast Cancer

Reflecting the intense interest in the role of miRNAs in breast cancer initiation, progression, and metastasis, there are currently ~3662 publications on miRNAs in breast cancer in PubMed. The reason behind this interest is that miRNAs are dysregulated in breast cancer (reviewed in [56,57,58,59,60,84]). Since each miRNA has the theoretical capacity to regulate multiple gene targets post-transcriptionally, alteration in the expression levels of just one miRNA has the potential to affect hundreds of target mRNAs. There are a number of programs to identify targets of a selected miRNA, and conversely, the miRNAs that may regulate a specific target gene (mRNA). Examples include miRTarBase, PicTar, miRanda, and MirAncesTar, although these programs predict false positives and false negatives (reviewed in [85]). Thus, experimental verification of the ability of a miRNA to downregulate a target requires cloning the 3′UTR of the target mRNA downstream of a luciferase reporter driven by an active promoter, e.g., CMV, mutating the miRNA recognition element in the reporter, and performing transient transfection assays in cells using the overexpression of wild-type miRNA and knockdown the miRNA expression. In addition, westerns and/or immunohistochemical staining (IHC) are needed to validate the decrease in authenticated target protein expression.

miRNAs are short (~22 nt) single-stranded RNAs that regulate mRNA stability and/or translation by base-pairing between the seed sequences at 5′ positions 2–7 or 2–8 of the miRNA with ~7 bp miRNA recognition elements (MREs) in the 3’UTR of their target mRNAs within the RNA-induced silencing complex (RISC) [86]. The current miRBase (release 22) reports 2654 mature miRNAs (http://www.mirbase.org/ [87]). Additional complexity is offered by the production of isomiRs that are generated from a single miRNA locus by template and non-template variants, and are differentially expressed in different types of human breast tumors [88]. The biogenesis of miRNAs has been extensively reviewed [28,89]. In summary, most miRNAs are transcribed as primary (pri)-miRNAs by RNA polymerase II, either cotranscribed within introns of host genes or as independent genes [90], and processed within the nucleus by the DROSHA-DGCR8 microprocessor complex [91]. DROSHA cleaves the hairpin-loop pri-miRNA yielding a 60–70 nt precursor (pre)-miRNA that is exported from the nucleus by the Exportin (XPO5) and Ran-GTP (RAN) or Exportin1 (XPO1, also called CRM1) [92]. The high/middle methylation of XPO5 was associated with reduced breast cancer risk, and XPO5 expression is increased in breast tumors [93]. An epidemiological study of key miRNA processing genes in the blood of male U.S. veterans reported that the DNA methylation of DROSHA and TNRC6B may play a role in early carcinogenesis [94]. DROSHA has tumor suppressor or oncogenic activity, depending on the type of tumor (reviewed in [95]). Single nucleotide polymorphisms (SNPs) in DROSHA are associated with increased breast cancer risk [96]. In the cytoplasm, the DICER-TRBP complex unwinds the double-stranded precursor miRNA (pre-miRNA) to allow the incorporation of one strand of the miRNA (called the guide strand) into the RISC complex that includes the catalytic Argonaut proteins, e.g., AGO2 [97]. The non-incorporated passenger strand of miRNA is degraded [98]. DICER is considered a tumor suppressor in breast cancer [95]. Increased DICER was associated with TAM resistance in metastatic breast tumors and tumor xenografts [99]. The high expression of AGO2 was reported to correlate with the luminal B subtype of breast cancer [100]. Interestingly, the overexpression of AGO2 in MCF-7 human breast cancer cells increased the expression of an ERα variant called ERα36, and stimulated E_2_-induced xenograft tumor growth in vivo in severe combined immunodeficiency (SCID)/Beige female mice [100]. This observation is of interest, since ERα36 plays a role in endocrine resistance and cancer stem cells (CSC) in breast cancer [101].

Next-generation sequencing approaches have identified ‘mitomiRs’ that function in mitochondria, whether by being imported from the repertoire of nuclear-encoded miRNAs or transcribed from the mitochondrial DNA (mtDNA) [102]. miRNAs may act as retrograde and anterograde signaling molecules between mitochondria and the nucleus to regulate energy homeostasis and apoptosis [102]. However, caution is needed, since the cytosolic contamination of mitochondrial extracts can lead to artifacts [102]. The role of mitomiRs in breast cancer is unknown. It is intriguing to note that PyMT mammary tumor metastasis was regulated by mtDNA in transgenic “mitochondrial-nuclear exchange (MNX) mice”, i.e., mice that had BL/6 mitochondria had longer tumor latency compared to those with FVB or BALB/c mitochondria, and mice with BALB/c mitochondria showed higher metastatic areas in lung [103,104]. At present, the precise mechanism(s) for this observation remain to be determined.

miRNAs that are overexpressed in breast tumor are called ‘oncomiRs’, since they promote carcinogenesis and progression by downregulating tumor suppressor genes. One example of an oncomiR in breast, as well as many cancers, is miR-21 [105,106,107,108,109,110,111,112,113,114,115]. Some examples of validated targets of miR-21 are indicated in Table 2. On the other hand, miRNAs that are downregulated in breast tumors are called tumor suppressor miRNAs. Notably, it is incorrect to assume that a miRNA is always an oncomiR or a tumor suppressor miRNA, since the activity of each miRNA depends on the cellular context, which includes mitigating factors such as lncRNAs and circRNAs (reviewed in [116]). It is important to note that many of the miRNAs that have been studied in breast cancer cell lines (reviewed in [53,59,117,118]) are not dysregulated in human breast tumors; thus, the focus here is on those identified in human breast tumors.

In breast cancer, miRNAs regulate genes involved in apoptosis, cell-signaling pathways (e.g., TGFβ [220], epithelial-to-mesenchymal transformation (EMT) [221], metastasis [222], the expression of ERα [117] and other nuclear receptors (NRs) [50]), regulation of the tumor microenvironment, and stemness, which includes transfer or exosomal miRNAs to adjacent normal fibroblasts forming cancer-associated fibroblasts (CAFs) (reviewed in [223]). Figure 1, Figure 2 and Figure 3 highlight the pathways of some of the miRNAs dysregulated in breast cancer, as summarized in Table 2 and Table 3. In addition, the lncRNAs that have been reported to ‘sponge’ some of the dysregulated miRNAs are shown as competing endogenous RNA (ceRNA).

Croce’s group first identified miRNAs dysregulated in breast cancer in 2005 [233]. There are numerous reviews covering 13 years of publications on the identity, targets, and regulation of miRNAs in breast cancer, and these references are not fully inclusive of all the published work [43,44,45,46,47,48,49,50,51,52,53,54,55,56,57,58,59,60,61,62]. The identity and possible roles of miRNA dysregulated in specific subtypes of breast cancer, including HER2+ [261,262], TNBC [263,264,265], and endocrine-resistant breast cancer [16,52,54,59,60] have been reviewed.

## 6. Long Non-Coding RNAs in Breast Cancer

The current GENECODE (version 28, GRCh38.p12) of the human genome includes 58,381 genes, 15,779 long non-coding RNAs (lncRNAs), and 1881 miRNAs (https://www.gencodegenes.org/stats/archive.html#a28). LncRNAs are defined as ncRNAs of >200 nucleotides [64]. They are transcribed from genomic DNA by RNA pol II, and are classified according to the genomic organization: (1) intergenic lncRNAs (lincRNAs) are transcribed between two protein-coding genes; (2) intronic lncRNAs are transcribed from the introns of protein-coding genes; (3) overlapping lncRNAs, which overlap protein-coding genes; and 4) antisense (as) lncRNAs, which are transcribed in a direction opposite to that of the protein-coding gene [65]. lncRNA expression is a low percent of total cellular RNA, i.e., ~0.03–0.20% [266]. Thus, while 15,779 lncRNAs have been identified, very few have been characterized with respect to cell type specificity and function. There is a need to develop tools for an improved analysis of differentially expressed lncRNAs. Since lncRNAs are low in expression, their expression levels are “very noisy”, which reflects their low counts in RNA-seq data [267]. Overall, quantitative examination of the identity of lncRNAs and their roles is needed, including in breast cancer.

In addition to nuclear-encoded lncRNAs, seven lncRNAs were identified as mitochondrial DNA transcripts mtDNA [268]. Two lncRNAs are encoded by mitochondrial D-loop regions: *MDL1* and *MDL1AS* [269]. There is evidence of mt-encoded lncRNAs in the nucleus, suggesting a potential role in retrograde signaling [268]; however, there are no reports on mtDNA-encoded lncRNAs in breast cancer.

Functionally, lncRNAs regulate transcription by associating with enhancer regions, in *cis*, i.e., at adjacent sites relative to their own transcription, or in *trans*, i.e., at more distal sites [270]. lncRNAs have heterogeneous complex 3D structures, which allows them to assume different shapes and interact with a wide variety of intracellular components. lncRNAs interact with the Mediator complex, forming loops between enhancer and promoter regions [266]. lncRNAs interact with proteins and other RNAs to influence their activities and cellular location. lncRNAs regulate development and differentiation, gene imprinting, and antiviral responses; and assist in chromatin modification, mRNA splicing, and protein stability [271]. Another function of lncRNAs is to act as ‘sponges’ (ceRNAs) for miRNAs, thus blocking the repressive activity of miRNAs for binding to the 3’UTR of their target transcripts. A network analysis of ncRNAs in cancer drug resistance-associated lncRNAs–miRNAs, TAM resistance (including lncRNAs *MALAT1* and *CCAT2*; miR-221, miR-222, miR-26a, miR29a, miR-29b), and Trastuzumab resistance (lncRNA *GAS5*, miR-16, and miR-155) has been described [272]. lncRNAs act as scaffolds, e.g., *HOTAIR* links the PRC2 and LSD1 histone-modifying complexes to promote histone H3K27 methylation and H3K4 demethylation to silence target genes and promote breast cancer metastasis [273].

The identity and roles of lncRNAs in breast cancer have been reviewed [28,52,274,275,276,277,278,279,280,281,282]. lncRNA expression profiles have been correlated with hormone status and intrinsic tumor type [283,284]. At present, there is no consistent identification of which lncRNAs in primary tumors are the best signature for predicting patient outcomes. For example, using four Gene Expression Omnibus (GEO) datasets (*n* = 473 breast cancer patients), one group identified a 12-lncRNA predictive signature for recurrence (*RP1-34M23.5*, *RP11-202K23.1*, *RP11-560G2.1*, *RP4-591L5.2*, *RP13-104F24.2*, *RP11-506D12.5*, *ERVH48-1*, *RP4-613B23.1*, *RP11-360F5.1*, *CTD-2031P19.5*, *RP11-247A12.8*, and *SNHG7*) [285], and another group using TCGA (*n* = 1064) identified a different set of three lncRNAs as prognostic markers (*CAT104*, *LINC01234*, and *STXBP5-AS1*) [286]. The differences may be the result of different platforms—the Affymetrix HG-U133 Plus 2.0 platform versus RNA-seq—in addition to the different tumors analyzed, the heterogeneity of breast tumors, the method of lncRNA identification and data interrogation, and, as stated earlier, the low level of lncRNA expression that confounds statistically relevant data interpretation. For this review, a selection of nuclear encoded lncRNAs that have been identified as dysregulated in breast tumors will be briefly summarized.

The lncRNA *RMRP* (RNA component of mitochondrial RNA processing endoribonuclease) binds RNA-binding proteins GRSF1, HUR, and PNPASE for transport into mitochondria, where *RMRP* plays a role in RNA processing and mtDNA replication [287]. Mutations were identified in the promoters of lncRNAs *RMRP* and nuclear paraspeckle assembly transcript 1 (*NEAT1*) that increased their expression in human breast tumors [288]. However, the expression of *NEAT1* showed no correlation with clinical gene signatures associated with higher grade, stage, metastasis, tumor aggression, or TAM resistance. Thus, the role of *NEAT1* in breast cancer is unclear [289]. Patients whose primary breast tumors showed a high expression of *NEAT1*, colon cancer associated transcript 2 (*CCAT2*), or metastasis associated lung adenocarcinoma transcript 1 (*MALAT1)* had shorter overall survival (OS) [282].

*NEAT1* is involved in the organization of nuclear architecture called paraspeckles for gene transcription and splicing [66]. Nuclear speckles are dynamic punctate compartments in the nucleus that contain components of the pre-mRNA spliceosome, including SRSFs, small nuclear ribonucleoproteins (snRNPs), RNA Pol II subunits, 3’ end processing proteins, m^6^A writers METTL3/14 and reader YTHDC1, and various protein kinases that regulate the pool of proteins in the speckles [290,291]. *NEAT1* was identified as an essential component of the FOXN3–SIN3A repressor complex, and the overexpression of *NEAT1* promoted EMT in ERα+ MCF-7 breast cancer cells and promoted the lung metastasis of MCF-7 when orthotopically implanted in the mammary fat pad, suggesting that *NEAT1* has oncogenic activity [292]. Bioinformatic analysis of sample-matched miRNA-seq and RNA-SeqV2 data of breast cancer from The Cancer Genome Atlas (TCGA) revealed that *NEAT1* was overexpressed in luminal A, luminal B, HER2+, and basal-like (TNBC) tumors [293]. Further, the authors identified a putative ceRNA network for *NEAT1*, as well as lncRNAs *OPI5-AS1* and *AC008124.1* in all breast tumors and each subtype [293]. *NEAT1* was also identified in a gene (*ESR1, DKC1*)–lncRNA (*TERC* and *TUG1*) interaction network in breast tumors from TCGA [294].

Another integrative analysis of RNA-seq data of ~1000 breast tumors in TCGA identified *GATA3-AS1* (ENSG00000197308), *RP11-279F6* (ENSG00000245750), and *AC017048* as highly expressed in ERα-positive versus ERα-negative breast tumors and normal breast tissue samples [295]. However, there are no confirming reports on these three lncRNAs or their function in breast cancer. Another analysis of TGCA RNA-seq data of human breast tumors identified a decreased expression of *LINC00092* and *C2orf71* as associated with poor prognosis, and identified a putative network of coexpression of *LINC00092* with mRNAs *RGMA* and *SFRP1* that were regulated by miR-449a and miR-452-5p [296]. However, neither of these miRNAs were dysregulated in breast tumors [61]. Further, *C2orf71* has been identified as a protein-coding gene *PCARE* in GeneCards.

The lncRNA *SRA1*, which is a steroid receptor RNA activator, was first identified in a complex with the coactivator SRC-1 (*NCOA1*) as an RNA coactivator that increased the transcriptional activity of NR, including ERα [297]. *SRA1* expression was higher in breast tumors compared with adjacent normal breast tissue [298]. *SRA1* is unique in that not only does it encode lncSRA1s of different length [299], but *SRA1* also encodes a protein: steroid receptor co-activator protein (SRAP) (reviewed in [300]). SRAP does not interact with the lncRNA *SRA1*, but it interacts with NRs: ERα, androgen receptor (AR), and glucocorticoid receptor (GR), and is involved in splicing and cell cycle regulation [301]. The lncRNA *SRA1* functions as a scaffold, and interacts with miRNA processing (DICER and TRBP) and RISC components, i.e., AGO2 and PACT; with other transcription factors, e.g., OCT4, NANOG, and FOXO1; and with chromatin modifiers and binding proteins, e.g., KMT2A (MLL1), KMT2D (MLL2), EZH2, and CTCF [301].

*MALAT1* (metastasis-associated lung adenocarcinoma transcript 1) is one of the best characterized lncRNAs with roles in neural development and function, retina, myogenesis, and vascular cell proliferation; it is dysregulated in cancers, including upregulation in breast tumors (reviewed in [302]). Serum levels of *MALAT1*, as examined by quantitative real-time PCR (qPCR), were higher in breast cancer patients (*n* = 157) than normal women (*n* = 107) [303]. *MALAT1* expression is associated with ERα+/PR+ breast tumors and with lower relapse-free survival (RFS) [282]. *MALAT1* staining was also higher in formalin-fixed paraffin embedded (FFPE) breast tumors than normal tissue [304]. *MALAT1* is oncogenic in breast cancer, and it upregulates the WNT/β-catenin (*CTNNB1*) pathway [305]. *MALAT1* mutations are frequent in breast tumors [306,307]. *MALAT1* acts as a ceRNA for miR-9, miR-133, miR-145, miR-195, miR-200s, miR-205, miR-206, and miR-503 (reviewed in [302]). *MALAT1* is targeted by interaction with miR-101, miR-125b, and miR-217. *MALAT1* acts as scaffold to position nuclear speckles at active gene loci. Capture hybridization analysis of RNA targets (CHART) revealed binding of *MALAT1* at actively transcribed loci (reviewed in [302]. Proteins interacting with *MALAT1* were identified by SILAC (stable isotope labeling with amino acids) labeling and LC-MS/MS proteomics in HEPG2 human hepatoma cells, and included proteins involved in RNA processing, splicing and gene transcription, and HNRNPAB [308]. Despite its lower expression in HER2+ and TNBC tumors, *MALAT1* expression was associated with decreased disease-specific survival in these patients [309]. Despite these studies implicating MALAT1 as oncogenic, a recent study in MMTV-PyMT;Malat1−/− mice demonstrated that *MALAT1* is a suppressor of lung metastasis in this model [251]. Notably, there was no difference in overall survival or tumor weight in the MMTV-PyMT;Malat1+/+ versus MMTV-PyMT;Malat1−/− mice and no difference in histological metrics of the tumors. The authors also reported that in TCGA RNA seq data, *MALAT1* was “underexpressed in human breast tumors compared with normal breast tissue”, and that lower *MALAT1* levels correlated with shorter distant metastasis-free survival in all breast cancer, as well as in luminal A and basal breast cancer [251]. Using chromatin isolation by RNA purification coupled to mass spectrometry (ChIRP-MS) studies, they demonstrated that *Malat1* sequestered the transcription factor TEAD, thus inhibiting its activity in mouse mammary tumors [251].

The expression of the lncRNA *CCAT2* (Colon Cancer Associated Transcript 2) is increased in breast tumors and breast cancer cells relative to normal breast tissue/cells [310,311]. Knockdown of *CCAT2* inhibited MCF-7 and MDA-MB-231 breast cancer cell proliferation and invasion in transwell migration assays by inhibiting WNT/β-catenin signaling [311]. *CCAT2* expression was higher in MDA-MB-231 and LCC9 TAM-resistant breast cancer cells derived from MCF-7 cells than in parental MCF-7 cells, and knockdown of *CCAT2* inhibited the activation of TGFβ signaling in LCC9 and MCF-7 cells [312]. *CCAT2* was reported to bind EZH2 and increased H3K27me3 in chromatin, thus repressing *CDKN2B* transcript and protein expression in MDA-MB-231 cells [313]. *CDKN2B* encodes a cyclin-dependent kinase inhibitor that interacts with CDK4 or CDK6, thus preventing interaction with cyclin D and inhibiting G1-S cell cycle progression [314]. Thus, the repression of CDKN2B would allow cell cycle progression. These results are as of yet unconfirmed in human breast tumors. The authors suggest the potential for therapeutic agents targeting cellular pathways linked to *MALAT1*.

Levels of the lncRNA *CRNDE* (colorectal neoplasia differentially expressed) were higher in breast tumors than normal breast and correlated with reduced OS [315]. *CRNDE* is a ceRNA for miR-136, resulting in the activation of WNT/β-catenin signaling in MDA-MB-231 cells [315]. Wnt signaling is a key driver of stem cells in embryonic and adult tissues and CSCs [316,317].

Analysis of TCGA breast tumor data identified higher expression of the lncRNA *MIAT* (myocardial infarction associated transcript) in breast tumors than normal breast tissue [184]. *MIAT* is a ceRNA for miR-155-5p, and knockdown of *MIAT* increased expression of the miR-155-5p target DUSP7 and inhibited MDA-MB-231 cell proliferation and xenograft tumor growth [184].

The lncRNA *LINC-ROR* was discovered in pluripotent stem cells, where it functions as a ceRNA for miR-145 to increase key pluripotency transcription factors OCT4, NANOG, and SOX2 [318]. *LINC-ROR* expression was higher in breast tumors and cell lines relative to normal breast tissue, and MCF-10A cells and the overexpression of *LINC-ROR* induced markers of EMT in MCF-10A cells [319]. High *LINC-ROR* expression in breast tumors was associated with reduced OS, and knockdown of *LINC-ROR* inhibited TGFβ signaling in MCF-7 and MDA-MB-231 cells [320]. Whether *LINC-ROR* is involved in the stimulation of CSC or metastasis remains to be examined.

Conversely, the expression of *MEG3* (Maternally Expressed 3) is lower in breast tumors than in normal breast tissues, and the expression level of *MEG3* was negatively correlated with histological tumor grade [282], and *MEG3* downregulation correlated with poor OS [321]. *MEG3* is a ceRNA for miR-421 in MDA-MB-231 TNBC and increased E-cadherin, while decreasing cell invasion in vitro [322]. The overexpression of *MEG3* in MDA-MB-231 cells suppressed xenograft tumor growth and angiogenesis, and reduced P-AKT, PCNA, and MMP-9 protein expression in the tumors formed in the mice [323]. Whether *MEG3* could be a potential therapy in breast cancer is not yet known, but is of interest in nervous system cancers [324].

lncRNAs continue to be discovered through the bioinformatic analysis of breast tumor data. For example, a recently published analysis of TCGA breast tumor identified putative driver lncRNAs: the amplification of *AC084809.2*, *RP11-108P20.3*, *ACOO5076.5*, *RP11-385J1.2*, *RP11-567G11.1*, *LINC00909, ATP1B3-AS1*, *MCCC1-AS1*, *YEATS2-AS1*, *TP53TG1*, and *SOX2-OT* [325]. As suggested previously, much more research is needed to understand the roles of lncRNAs in breast cancer initiation, progression, and metastasis.

## 7. Micro RNAs and Long Non-Coding RNAs in Extracellular Vesicles and Exosomes

Exosomes are a type of extracellular vesicle (EV) that are ~50–140 nm in size, endosome-derived, and secreted by most cells under normal and disease states [326]. Surface markers on exosomes include CD9, CD63, CD81, LAMP1, and TSG101 [327]. Microvesicles (MVs, also called ‘ectosomes’) are EVs that bud from the plasma membrane surface, but many people refer to MVs and exosomes interchangeably [328]. Exosomes contain ncRNA (circRNA, miRNA, lncRNA) in addition to dsDNA, mRNA, proteins, lipids, DICER, and TRBP, as well as AGO2, and process pre-miRNAs to mature miRNAs [327]. A web-based database exoRBase is a repository of 15,501 lncRNA, 58,330 circRNA, and 18,333 mRNA [329]. Exosome levels are higher in the serum of breast cancer patients compared to normal subjects [330]. EVs deliver cargo, including ncRNA, lipids, mtDNA, and proteins, between cells and play roles in the context of tumor growth and the stimulation of metastasis while suppressing immune detection [328,331]. EVs, and exosomes in particular, are considered of great promise as ‘liquid biopsy’ and biomarkers in cancer detection and monitoring therapeutic response [332]. Examples of exosomal miRNAs that are released by breast cancer cells and found in higher amount in the serum of breast cancer patients than controls include miR-21, miR-195, miR-484, and miR 1246 [332]. However, there is concern about the manner of sample collection and variability in content of EVs, and exosomes due to different methods of isolation that are yet to be standardized for clinical lab testing [328].

The role of exosomal miRNAs in breast cancer has been reviewed [60,333,334]. Exosomes are secreted by cancer cells and by cancer-associated fibroblasts, and can be taken up by neighboring cells in the tumor environment, including NK cells and T lymphocytes, as well as distant cells after traveling in blood [60]. Examples of exosomal miRNAs and their targets in breast cancer are shown in Figure 4. Exosomes secreted from TAM-resistant MCF-7 cells in vitro were shown to confer TAM resistance to parental MCF-7 cells, in part by delivering miR-222 and miR-221 that repress *ESR1* (ERα) translation [335]. Exosomes from TAM-resistant LCC2 cells, derived from MCF-7 cells, had higher levels of the lncRNA UCA1 than parental MCF-7 cells, and the incubation of MCF-7 cells with exosomes from LCC2 cells conferred TAM-resistance [336]. We reported that miR-29b-1/a directly targets ATP synthase subunit genes *ATP5G1* and *ATPIF1* and inhibit OCR and decrease ATP in MCF-7 and LCC9 breast cancer cells [337]. UCA1 activates AKT-mTOR signaling [338] and is a ceRNA for miR-18a, thus derepressing HIF-1α [137]. The lncRNA MALAT1 is found in exosomes from breast cancer patients [339]. MALAT1 is known to activate WNT–β-catenin signaling, and is oncogenic in breast cancer (reviewed in [28]). Most studies have characterized the activities of miRNAs and lncRNAs in cancer cell exosomes in vitro; thus, research is needed to validate the role of these miRNAs and lncRNAs in breast cancer metastasis in vivo.

## 8. Concluding Considerations

ncRNAs are dysregulated in breast cancer and miRNAs, noted for their stability, and lncRNAs are being investigated as biomarkers of disease pathology, prognostic indicators, and potential therapeutic targets, as well as modalities to block cancer progression and metastasis. We know much more about miRNAs than lncRNAs in human breast tumors. Our understanding of the networks between miRNAs, lncRNAs, their interacting partners, and targets is expanding. However, individual and combinations of miRNAs and lncRNAs have cell-specific activities that are not fully understood, nor is their interaction with the immune system, microbiome, microenvironment, hormonal milieu, or metabolome elucidated. Further investigation is needed to bring studies on miRNAs and lncRNAs into clinical practice for diagnosis, prognosis, and therapeutics in breast cancer patients.

## Figures and Tables

**Figure 1 ncrna-04-00040-f001:**
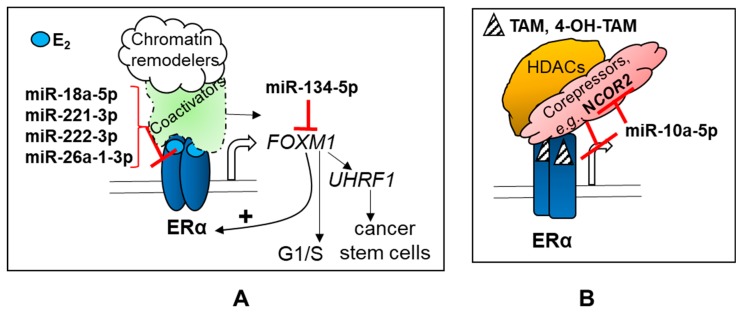
miRNAs regulating ERα transcriptional activity. (**A**) ERα is directly targeted by the indicated miRNAs that are increased in breast tumors (Table 2). E_2_-liganded ERα recruits coactivators and chromatin remodeling complexes to increase RNA pol II transcription at target genes. E_2_–ERα increases the transcription of *FOXM1*, which, in turn as a transcription factor, increases the transcription of ERα, including a number of genes for cell cycle progression [224], and *UHRF1*, which is a key regulator of DNA methylation that is involved in the self-renewal and differentiation of cancer stem cells [225]. (**B**) The selective ER modulator (SERM) tamoxifen is metabolized to 4-hydroxytamoxifen, which binds ERα and alters its conformation, thus inhibiting coactivator recruitment, and instead allowing interaction of the 4-OHT-bound-ERα with corepressors, including NCOR2, which recruits histone deacetylase complex (HDAC) complexes to inhibit target gene transcription in breast tumors. NCOR2 is a target of miR-10a-5p (Table 2).

**Figure 2 ncrna-04-00040-f002:**
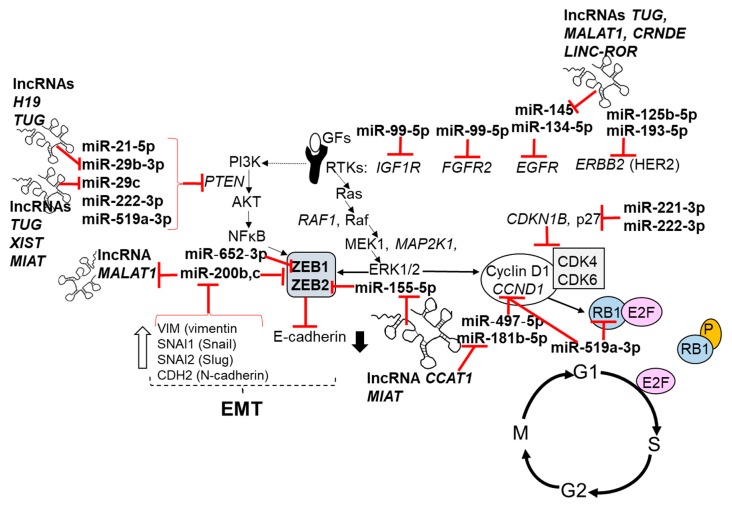
Breast cancer dysregulated miRNAs and lncRNAs as ceRNAs in cell signaling, cell cycle, and EMT. Shown are validated targets of some miRNAs dysregulated in human tumors and lncRNAs that at as ceRNAs for the indicated miRNAs.

**Figure 3 ncrna-04-00040-f003:**
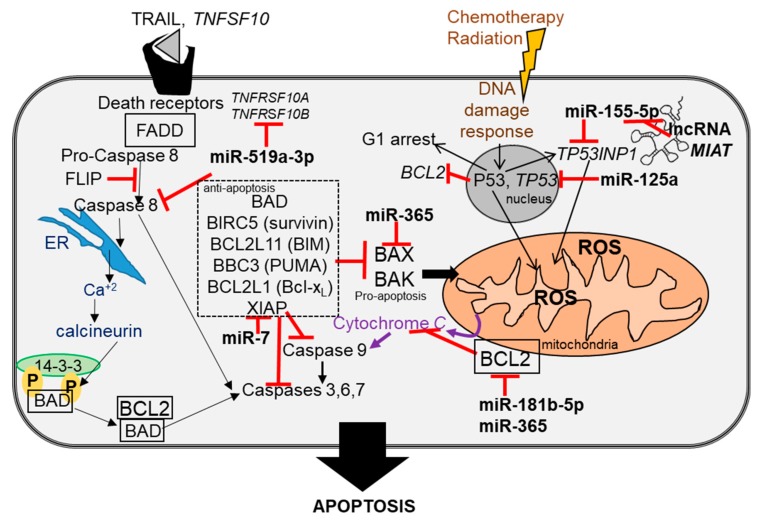
Breast cancer dysregulated miRNAs in apoptosis. Shown in abbreviated form are key regulators in the intrinsic and extrinsic pathways of apoptosis and their regulation by miRNAs that are dysregulated in breast tumors (Table 2 and Table 3). The lncRNA *MIAT* is a ceRNA for miR-155-5p (Table 2).

**Figure 4 ncrna-04-00040-f004:**
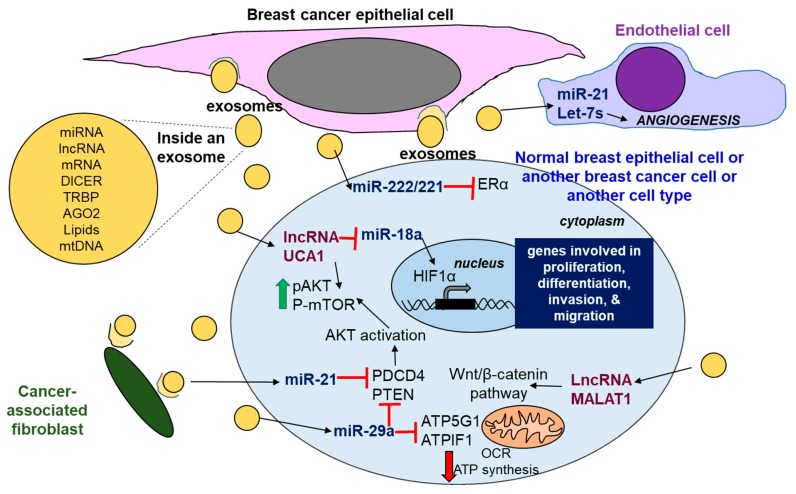
**Exosomal transfer of miRNAs and lncRNAs in breast cancer.** Exosomes released from breast cancer cells and cancer-associated fibroblasts into the extracellular compartment contain ncRNAs, mRNAs, mtDNA, proteins, and lipids. Exosomes can deliver their contents to adjacent cells or cells at a distance. Examples of miRNAs and lncRNAs in breast cancer exosomes and their known roles in breast cancer are shown.

**Table 1 ncrna-04-00040-t001:** Regulatory non-coding RNAs (ncRNAs) in breast cancer. Abbreviations: BCa (breast cancer), ceRNA: competing endogenous RNA; TNBC: triple negative breast cancer, ssRNA: single-stranded RNA.

ncRNA	Description, Size, Cellular Location, Function	Examples in Breast Cancer
Circular RNAs (circRNAs)	Circular RNAs: Four types: circular RNAs from introns, exonic circRNAs, exon-intron circRNAs (EIciRNA), and intergenic circRNAs [29] Size range from a few hundred to >1000 nt Nucleus < cytoplasm Nuclear export is active and size-dependent, involving DDX39B and DDX39A [30] Generally formed by alternative splicing of pre-mRNA, in which an upstream splice acceptor is joined to a downstream splice donor in a process known as ‘backsplicing’ [31] Expressed in thousands of human genes Stable: half-life >48 h Act as miRNA ‘sponges’ (ceRNA), interact with RNA binding proteins, can be positive regulators of their parental genes Can be translated [32]	Tumor-specific circRNAs were identified in human breast tumors and BCa cells (BT-20, BT-474, MCF-7, MDA-MB-231, MDA-MB-468, T-47D, and ZR-75-1) [33] circ-ABCB10 was upregulated in BCa tumors and acted as a ceRNA for miR-1271 [34] hsa_circ_0001982 was overexpressed in BCa tissues and cell lines, and acted as a ceRNA for miR-143 [35]. circGFRA1 was upregulated in TNBC and acted as a ceRNA for miR-34a Analysis using CircHunter and HashCirc recently identified circRNAs in MCF-7 BCa cells [36] circIRAK3, was increased in TNBC cells (MDA-MB-231, MDA-MB-157, HCC70, HCC1806, HCC1937) and acts as a ceRNA for miR-3607 [37] circ_0008039 was upregulated in BCa tissues, and acted as a ceRNA for miR-432-5 [38]
PIWI-interacting RNAs (piRNA)	24–31 nt ssRNAs [39] Derived from piRNA clusters; do not require DICER Nuclear and cytoplasmic Bind PIWI subfamily of Argonaut proteins Involved in gene silencing [40]	piRNAs: DQ596670, DQ598183, DQ597341, DQ598252, and DQ596311 were underexpressed; DQ598677, DQ597960, and DQ570994 overexpressed in BCa tissues compared to normal breast tissue [41]. piR-021285 was identified as a potential modulator of BCa invasiveness, which is a function linked to piR-021285-dependent 5′ UTR/first exon methylation of the pro-invasive *ARHGAP11A* gene [42].
MicroRNAs (miRNA)	~22 nt ssRNA Nucleus and cytoplasm Forms complimentary base-pairs with the 3′ untranslated region (UTR) of target mRNAs within the RNA-induced silencing comples (RISC) complex to inhibit translation and/or stimulate mRNA transcript degradation most are considered highly stable	Reviewed in [43,44,45,46,47,48,49,50,51,52,53,54,55,56,57,58,59,60,61,62].
lncRNA	200 nt Transcribed by RNA pol II: intergenic (lincRNA), intronic, antisense, and overlapping long non-coding RNAs (lncRNAs) 5′ cap and polyadenylated Tissue-specific expression Gene loci marked with H3K4me3 at the promoter and H3K36me throughout the transcript body Roles in transcriptional, splicing, translation, intracellular protein localization, nuclear architecture, cell cycle, cancer stem cells, and apoptosis	Reviewed in [63,64,65,66]

**Table 2 ncrna-04-00040-t002:** Examples of miRNAs upregulated in breast tumors, patient plasma, and breast cancer cell lines with their authenticated targets. miRTarBase was used to identify validated targets [119], i.e., those not identified by reference number. Abbreviations: AI = aromatase inhibitors, BCa = breast cancer; ceRNA = competing endogenous RNA (‘miRNA sponge’), CSCs = cancer stem cells, DCIS = ductal carcinoma in situ, EMT = epithelial-to-mesenchymal transformation, ERα = estrogen receptor α, HN = healthy normal, PR = progesterone receptor, TAM = tamoxifen, TNBC = triple negative breast cancer, TAM = tamoxifen.

miRNA Increased in BrCa	Examples of Validated Target (s)	Pathway (s)	Comments
miR-7	*XIAP* [120]	TRAIL-induced apoptosis [121]	Higher in DCIS than HN [122]; High in BCa tissues [56]; miR-7 expression was negatively correlated with the stage, grade, and survival of BCa patients [123].
miR-10a-5p	*ACTG1* [124]; *BCL6* [125]; *CHL1* [126]; *NCOR2* [127]; *MMP14* and *SKA1* [128]; *YAP1* [129]; lncRNAs *TUSC7* and *RP11-838N2.4* act as ceRNAs for miR-10a in HCC [130] and glioblastoma [131] cells	PI3K/AKT/mTOR pathway [132]	High expression was predictive of tumor relapse in TAM-treated ER+ postmenopausal BCa patients [133,134]; Low miR-10a-5p correlated with reduced relapse-free survival in BCa [16].
miR-18a-5p	*ESR1*/ERα [135] *SREBP1* [136] lncRNA *UCA1* is a ceRNA miR-18a in BCa cells [137]		Higher in ERα-breast tumors [138]; higher in DCIS than HN [122]; Higher in metastatic BCa cells where it decreased *ECAD* (E-cadherrin) and increased EMT and metastasis of xenografted MDA-MB-231 TNBC cells [136];
miR-21-5p	*PTEN*, *PDCD4* [106,139]; *NFIB* [122]; *RASA1* and *RASA2* [140]; *BTG2*, *FBXO11*, *MARCKS*, *RECK*, and *TPM1* [141]; *TIMP3* [115] miR-21 negatively regulates lncRNAs *GAS5* and *CASC2* [28]	PI3K-AKT signaling, apoptosis	Consistently increased in breast tumors and in plasma from BCa patients [56]. High in breast tumors; high miR-21 correlates with lymph node status and tumor stage [107]. A meta-analysis of serum/plasma miR-21 in 438 BCa patients and 228 healthy controls concluded that increased miR-21 is a potential biomarker for BCa with a sensitivity of 0.79 [142].
miR-26a-1-3p	*ESR1* [143]; *CHD1*, *GREB1*, and *KPNA2* [144] *CDC2*, *CCNE1* [145]; *EZH2* [146]	Apoptosis in cancer cells [146]	Higher expression in primary breast tumors was associated with clinical benefit of tamoxifen [16].
miR-29b-3p, miR-29c	*DICER*, *TTP*, *PTEN*, *ARP1B1*, *KLF4*, *MYP*, *ANGPTL4*, *LOX*, *MMP*, *PDFGC*, *VEGFA*, *ADAM12*, *SERPINH1* (reviewed in [60,147]). LncRNAs: H19 targets miR-29b-3p [148,149]; TUG targets miR-29b and miR-29c [150]; *XIST* targets miR-29c [151,152]; *MIAT* targets miR-29c [153].	Have both tumor suppressor and oncomiR roles [154]	Upregulated in BCa tissues [56].
miR-30c-5p	*CTGF* [155]; *BCL9* [156]; *HOXA1* [157]; *SRSF1* [158]; *KRAS* [159]; *CHD7* and *TNRC6A* [160] lncRNA *AK017368* acts as a ceRNA for miR-30c in skeletal muscle cells [161]	Proliferation, apoptosis, differentiation [162]. Oncogene-induced senescence: a key tumor-suppressing mechanism [160]	High miR-30c-5p in primary tumors associated with clinical benefit of tamoxifen treatment [16]. Low miR-30 family expression in breast tumors was associated with poor relapse-free survival and bone metastasis [163].
miR-96-5p	*MTOR* and *RPS6KB1* (also called S6K1) in TNBC [164]	Insulin signaling in non-small cell lung cander (NSCLC) [165]	Upregulated in BCa samples [166].
miR-125b-5p	*ERBB2*, *ERBB3* [167]; *ETS1* [168]	Epidermal growth factor receptor (EGFR) signaling Resistance to TAM and AIs [59]	In ER+/PR+ patients, high miR-125b-5p correlated with earlier relapse [134,169]. Also increased in blood plasma from BCa patients [170,171].
miR-134-5p	*EGFR* [172], *FOXM1*, *KRAS*, *STAT4B*, *ERBB2*	Cell proliferation, apoptosis, invasion, metastasis, drug resistance; however, it also acts as a tumor suppressor miRNA by targeting STAT4B, KRAS, and the Notch signaling pathway [173]. FOXM1 increases transcription of genes for G1/S transition, promotes CSCs and endocrine resistance [174].	Increased in circulating plasma from BCa patients [175]. Encoded in the DLK1-DIO3 genomic region, located on 14q32 that contains the paternally expressed imprinted genes *DLK1*, *RTL1*, and *DIO3*, and the maternally expressed imprinted genes *MEG3*, *MEG8*, and as RTL1, two lncRNAs, and 53 miRNAs [176]. FOXM1 transcriptionally increased E_2_–ERα in BCa cells, and regulation is reciprocal [177].
miR-155-5p	*TERF1* [178]. *TP53INP1* [179]; *ZEB2* [180]; lncRNA H19 [181] lncRNAs acting as ceRNA for miR-155 include *CCAT1* [182]; *MALAT1* [183], *MIAT* [184], and *CCAT1* [185] in cancer cells	EMT and metastasis [180]	Upregulated in BCa tissues [56]; higher in the serum of BCa patients than healthy women [186].
miR-181b-5p	*CCND1*, *CBX7*, *BCL2*, *HMGA2*, *TP53* [187]; *DAX1* [141] lncRNA *CCAT1* acts as a ceRNA for miR-181b in glioma cells [188]	Growth factor signaling [59]	Higher in DCIS than HN [122]; levels of miR-181b decline in serum after surgical removal of breast tumors [189].
miR-181b-3p	*YWHAG* [190]	EMT [190]	Higher in metastatic BCa lines versus MCF-7 and T47D [190]
miR-185-5p	*VEGFA* [191]; *E2F6* and *DNMT1* [192]	Upregulated by tumor and metastasis suppressor *PEBP1* (also called RKIP) in BCa cells [193]; apoptosis [194];	Reduced in BCa tissues [193].
miR-193a-3p	*DDAH1* [195]; *WT1* [196]	Cell growth [196] miR-193a-5p was decreased, but no significant differences in miR-193a-3p in BCa [197].	Highly expressed in breast tumors [198]; downregulated in BCa tumors [196]
miR-210-3p	*ISCU* and *COX10* [199]; *FGFRL1*, *RAD52*; *BDNF*, *PTPN1*, *ISCU*, *NCAM1*, and the lncRNA *XIST* [200]	Cell proliferation, migration, and invasion	High miR-210 was associated with lower relapse-free survival [201]. Induced by hypoxia [202] Increased in cell line models of aromatase resistance [172]. Consistently increased in breast tumors and plasma from BCa patients [56].
miR-221-3p	*ESR1*/ERα [203], *CDKN1B*, *FOXO3*, *KIT*, *TIMP3*, *BRAP*, *ARIH2*, *FOS*, *ICAM1*	ERα regulation of gene transcription in BCa [204,205]. Both miR-221 and miR-222 are involved in regulating adherens junction, PI3K and MAPK signaling, transforming growth factor β (TGFβ)signaling, apoptosis, and cell cycle [206].	Increased in tumors of patients who develop tamoxifen resistance [207].
miR-222-3p	*ESR1*/ERα [203], *TIMP3* [208], *STAT5A* [209], *MMP1* [210], *FOXO3*, *FOX*, *PTEN*, *KIT*, *SOCS1* and *CDKN1B* [211]	ERα regulation of gene transcription in BCa [204]. Promotes S-phase entry, EMT, and TAM resistance [172].	Increased in breast tumors [212]. High expression of miR-222 was associated with short relapse-free time in ER+/PR+ BCa patients [134].
miR-324-5p	*SMO* [213]	Suppressed invasion of MDA-MB-231 cells [214]	Higher in DCIS than HN [122];
miR-365	*BCL2* [215]; *SHC1* and *BAX* [216]	Higher circulating levels in plasma predicted decreased OS in metastatic BCa patients [217].	Higher in DCIS than HN [122];
miR-519a-3p	*CDKN1A*, *RB1*, and *PTEN* [218]; *TNFRSF10B* (TRAIL-R2) and *CASP8* [219]	TRAIL-induced apoptosis [219].	High expression correlated with lower disease free survival in ER+ patients, not ER- patients [218]. TAM resistance in MCF-7 cells: transient transfection of MCF-7 cells with a miR-519a mimic resulted in TAM resistance; conversely, the transfection of TAM-resistant MCF-7 cells with a miR-519a inhibitor restored TAM growth inhibition [218].

**Table 3 ncrna-04-00040-t003:** Examples of miRNAs downregulated in breast tumors, patient plasma, and breast cancer cell lines with their authenticated targets. miRTarBase was used to identify validated targets not identified by reference number.

miRNA Decreased in BrCa	Examples of Validated Target (s)	Pathway (s)	Comments
miR-99a-5p	*IGF1R*, *AKT1* [226]; *HOXA1* [227]; *RAVER2*, *FGFR2*, *IGF1R*, *MTOR*, *AGO2*	TGFβ pathway [228]; mTOR signaling pathway [229]	Downregulated in BCa tissues [56]; downregulated in DCIS [230]; low serum miR-99a is a poor prognostic indicator in BCa correlating with lymph node metastasis, and distant metastasis [231].
miR-125a-5p	*CDKN1*, *NTRK3*, *TP53*, *VEGFA*, *ERBB2*, *ERBB3*, *BAK1*, *KLF13*, *ARID3B*, *ELAV1* lncRNAs that act as ceRNAs for miR-125a-5p include *HOTAIR*, *ANRIL*, and *HOXA11-AS* (reviewed in [28])	NFκB pathway [232]	Lower in breast tumors [233].
miR-127	*PRDM1* [234] lncRNA *MEG3* acts as a ceRNA for miR-127 in osteosarcoma cells [235]	p53 transactivates miR-127 leading to the inhibition of *MMP13* translation, whereas c-Jun (activated by TGFβ) inhibits miR-127 transcription [236]	Lower in DCIS than histologically normal tissue [122].
miR-139-5p	*POLQ*, *TOP1*, *TOP2A*, *RAD54L*, and *XRCC5* [237]; RUNX1 [238]	EMT [239]	miR-139-5p was downregulated in BCa tissues [56].
miR-143	*MAPK1* [240]; *DNMT3A* [241]; *CIAPIN1* [242]; *BCL2* [243];	RAS signaling in basal-like BCa [244].	Downregulated in BCa tissues [56].
miR-145	*RASA1*, *MEKK*, *EGFR* [245]; *TGFNR2* and *SMAD3* [246] lncRNAs act as ceRNAs for miR-145-5p: *TUG1* and *MALAT1* [246,247]; *CRNDE* in gastric cancer cells [248], *LINC-ROR* in gastric cancer cells [249], and *PCAT1* in prostate cancer cells [250]	RAS signaling in basal-like BCa [244]. TEAD-YAP transcriptional pathway, including *VEGFA* and *ITGB4* [251].	Lower in breast tumors [233]; downregulated in BCa tissues [56].
miR-193a-5p	*ERBB2* (HER2) [252]; *WT1*, *SRSF2*, *HIC2*, *HOXC9*, *PSEN1*, *LOXL4*, *ING5*, *c-KIT*, *PLAU*, and *MCL1* [196]		Downregulated in DCIS [230].
miR-378a-3p	*GLI3* [253]; *GOLT1A* [254]		Downregulated in DCIS [230].
miR-497-5p	*BCL2* and *CCND1* [255]; *KCNN4* [256]		Downregulated by methylation in breast tumors [257]; Downregulated in DCIS [230].
miR-652-3p	*ZEB1* [258]	EMT [259]	Downregulated in DCIS [230] Lower in serum of BCa patients [260].

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
