# Peer review of "Non-Coding RNAs in Breast Cancer: Intracellular and Intercellular Communication"

_ncrna, 2018, doi:10.3390/ncrna4040040_

Round 1
Reviewer 1 Report
This is an excellent and very comprehensive review that is important for the field. There were some minor editorial suggestions that have been included in the manuscript for the authors consideration. Other things that should be addressed:
There are way too many references. References are incomplete and not in the proper format. Specific references noted to be incomplete include: 2, 4, 53, 57, 127, 141, 143, 156, 205, 217, 222, 229, 236, 270, 276, 299, 322 and 323
Section 6 and 7 should have the order reversed.
We would consider removing section 8, it adds little and does not tie into the review well.
Other comments are mentioned in the attached PDF.

Author Response
Reviewer 1: This is an excellent and very comprehensive review that is important for the field. There were some minor editorial suggestions that have been included in the manuscript for the authors consideration. Other things that should be addressed:
There are way too many references.
I acknowledge that there are > 300 references; however, while ~ 10 sere eliminated by removing Section 8 as Reviewer 1 suggested to address concerns from Reviewer 2, I added more.
References are incomplete and not in the proper format. Specific references noted to be incomplete include: 2, 4, 53, 57, 127, 141, 143, 156, 205, 217, 222, 229, 236, 270, 276, 299, 322 and 323
I have corrected references 2, 4, 143, 156, 205, 217, 222, 229, 270, 276, 299, and 323.
However, Ref. #53, 57, 127, 141, 236, 322 (I hand-corrected title from what is in PubMed) are still Epublished ahead of print, so volume and pg. # are not yet available for inclusion.
Section 6 and 7 should have the order reversed. I have reversed the order as suggested by Reviewer 1 for Sections 6 and 7.
We would consider removing section 8, it adds little and does not tie into the review well. As suggested by Reviewer 1, I removed section 8.
Other comments are mentioned in the attached PDF. I appreciate the detection and correction of a small number of my typographical errors.
Reviewer 2 Report
The present paper presents an overview upon ncRNAs in breast cancer with possible relevance in diagnosis, prognosis and also therapeutic strategies for this malignancy. The article has a good structure and can be of interest for readers that are aiming to get an overview about the non-coding communication in breast cancer. However, there are still some improvements that can be made, and we cannot recommend the publication until the authors reach the following concerns:
- Non-coding RNAs have been intensely studied in the filed of cancer and not only and DNA and RNA-Seq technologies contributed greatly to this research; However, there is a misconception about the actual role of these sequences, where much of the studies are just profiling the samples without considering the actual level of expression. Much of the ncRNAs are expressed as such a low amount that, despite their possible target genes, they are unable to perform the predicted function. These aspects have been reviewed by Alexander F. Palazzo and Eliza S. Lee (https://www.ncbi.nlm.nih.gov/pmc/articles/PMC4306305/) and we suggest that at least a brief statement about this subject should be introduced in the article.
- The chapter about snRNAs and snoRNAs in breast cancer is significantly under approached in comparison to others; considering that there are currently an extensive numbers of reviews about miRNAs or lncRNAs in breast cancer, authors should consider adding more information to this chapter in order to increase the value of the manuscript
- The same is the case of circRNAs chapter; at the current time these transcripts are highly studies to their heterogenous function within the cell, including breast cancer cells. Authors should consider adding more information about the role of these ncRNAs in breast cancer.
- The subject about “mitomiRs” is interesting and not so extensively approached in review articles; authors should consider including more data about miRNAs within mitochondria and their role in breast cancer
- Figure 2 is complex and informative; however considering that the main pathology discussed within the review is breast cancer, authors should focus only on the miRNAs/mechanisms demonstrated in this malignancy and not include data from “some miRNAs dysregulated in human tumors and lncRNAs”
- Figure 3 should be more organized
- Chapter 6 – miRNAs and lncRNAs in extracellular vesicles should be added after the presentation of lncRNAs (Chapter 7)
- Authors should complete the chapter about exosomes with an informative figure about their signaling roles between cells in order to help non specific specialized readers in understanding this idea (at the current time, the chapter is quite short, so a figure could help)
- Circular RNAs therapeutic role in sponging miRNA should be included in the therapeutic chapter
- English proofreading should be made as there are several grammatical and expression errors
Author Response
Reviewer 2: The present paper presents an overview upon ncRNAs in breast cancer with possible relevance in diagnosis, prognosis and also therapeutic strategies for this malignancy. The article has a good structure and can be of interest for readers that are aiming to get an overview about the non-coding communication in breast cancer. However, there are still some improvements that can be made, and we cannot recommend the publication until the authors reach the following concerns:
- Non-coding RNAs have been intensely studied in the filed of cancer and not only and DNA and RNA-Seq technologies contributed greatly to this research; However, there is a misconception about the actual role of these sequences, where much of the studies are just profiling the samples without considering the actual level of expression. Much of the ncRNAs are expressed as such a low amount that, despite their possible target genes, they are unable to perform the predicted function. These aspects have been reviewed by Alexander F. Palazzo and Eliza S. Lee (https://www.ncbi.nlm.nih.gov/pmc/articles/PMC4306305/) and we suggest that at least a brief statement about this subject should be introduced in the article.
I fully agree with the Reviewer’s point and appreciate the reference. I have added a new sentence on page 2 to address the fact that the levels of ncRNA expression, their post-transcriptional modification (particularly lncRNAs), and their subcellular distribution are important to consider in assigning potential function. In fact, in agreement with Palazzo and Lee’s comments in the paper https://www.ncbi.nlm.nih.gov/pmc/articles/PMC4306305/, it is ultimately critical to examine the biological function of each identified ncRNA on a case-by-case basis.
- The chapter about snRNAs and snoRNAs in breast cancer is significantly under approached in comparison to others; considering that there are currently an extensive numbers of reviews about miRNAs or lncRNAs in breast cancer, authors should consider adding more information to this chapter in order to increase the value of the manuscript
As suggested by Reviewer 2, I have added additional information and references to Section 2.
- The same is the case of circRNAs chapter; at the current time these transcripts are highly studies to their heterogenous function within the cell, including breast cancer cells. Authors should consider adding more information about the role of these ncRNAs in breast cancer.
As suggested by Reviewer 2, I have added additional information and references to Section 2.
- The subject about “mitomiRs” is interesting and not so extensively approached in review articles; authors should consider including more data about miRNAs within mitochondria and their role in breast cancer
I agree that the potential role of mitomiRs is indeed interesting; however, I did not locate any papers related to breast cancer in PubMed. To address this comment, I added 2 sentences relating to the work of Danny Welch showing that the strain of mouse mitochondria impact PyMT mammary tumor latency and metastasis in mice at the end or section 5.
- Figure 2 is complex and informative; however considering that the main pathology discussed within the review is breast cancer, authors should focus only on the miRNAs/mechanisms demonstrated in this malignancy and not include data from “some miRNAs dysregulated in human tumors and lncRNAs”
In fact, Figure 2 does specifically show dysregulated miRNAs and lncRNAs as ceRNAs in cell signaling, cell cycle, and EMT in breast cancer. The Figure legend needs correction to add the word breast before tumors to make that clear: “Shown are validated targets of miRNAs dysregulated in human breast tumors and lncRNAs that at as ceRNAs for the indicated miRNAs.”
- Figure 3 should be more organized
I have revised and replaced Figure 3. I did not use Tracker when I pasted it into the manuscript.
- Chapter 6 – miRNAs and lncRNAs in extracellular vesicles should be added after the presentation of lncRNAs (Chapter 7)
I have reversed the order as suggested by Reviewer 2 for Sections 6 and 7.
- Authors should complete the chapter about exosomes with an informative figure about their signaling roles between cells in order to help non specific specialized readers in understanding this idea (at the current time, the chapter is quite short, so a figure could help)
I have added new information, references, and a new Figure (Figure 4) to address this comment.
- Circular RNAs therapeutic role in sponging miRNA should be included in the therapeutic chapter
Reviewer 1 suggested deleting this section, so I did.
- English proofreading should be made as there are several grammatical and expression errors
I apologize for various typographical errors that I have fixed in this revision.
Round 2
Reviewer 2 Report
The manuscript can be accepted after all the improvements done.